# Hydrothermal Synthesis of Aqueous-Soluble Copper Indium Sulfide Nanocrystals and Their Use in Quantum Dot Sensitized Solar Cells

**DOI:** 10.3390/nano10071252

**Published:** 2020-06-28

**Authors:** Calink I. L. Santos, Wagner S. Machado, Karl David Wegner, Leiriana A. P. Gontijo, Jefferson Bettini, Marco A. Schiavon, Peter Reiss, Dmitry Aldakov

**Affiliations:** 1Grupo de Pesquisa em Química de Materiais (GPQM), Departamento de Ciências Naturais, Universidade Federal de São João del-Rei, Campus Dom Bosco, Praça Dom Helvécio, 74, CEP 36301-160 São João del-Rei, MG, Brazil; calinkindiaradc@hotmail.com (C.I.L.S.); wagner@ufsj.edu.br (W.S.M.); leiriana@hotmail.com (L.A.P.G.); schiavon@ufsj.edu.br (M.A.S.); 2Univ. Grenoble Alpes, CEA, CNRS, IRIG-SyMMES, STEP, 38000 Grenoble, France; karl-david.wegner@bam.de (K.D.W.); peter.reiss@cea.fr (P.R.); 3Laboratório Nacional de Nanotecnologia, Centro Nacional de Pesquisa em Energia e Materiais, CEP 13083-970 Campinas-SP, Brazil; jefferson.bettini@lnnano.cnpem.br

**Keywords:** chalcopyrite, CIS, aqueous quantum dots, fractionation, quantum dot sensitized solar cells

## Abstract

A facile hydrothermal method to synthesize water-soluble copper indium sulfide (CIS) nanocrystals (NCs) at 150 °C is presented. The obtained samples exhibited three distinct photoluminescence peaks in the red, green and blue spectral regions, corresponding to three size fractions, which could be separated by means of size-selective precipitation. While the red and green emitting fractions consist of 4.5 and 2.5 nm CIS NCs, the blue fraction was identified as in situ formed carbon nanodots showing excitation wavelength dependent emission. When used as light absorbers in quantum dot sensitized solar cells, the individual green and red fractions yielded power conversion efficiencies of 2.9% and 2.6%, respectively. With the unfractionated samples, the efficiency values approaching 5% were obtained. This improvement was mainly due to a significantly enhanced photocurrent arising from complementary panchromatic absorption.

## 1. Introduction

In the last decade, colloidal ternary chalcopyrite nanocrystals (NCs), such as CuInS_2_ (CIS) and CuInSe_2_, emerged as promising materials for photovoltaic applications. They combine several appealing features such as the absence of toxic heavy elements like Cd or Pb, high absorption coefficients with an appropriate band gap (1.5 eV for CIS) and long-lived excitonic states allowing for efficient harvesting of sunlight and charge transfer, respectively [1,2,3]. Probably the most adapted architecture for applications of ternary NCs in photovoltaics is the quantum dot sensitized solar cell (QDSSC), an approach which currently holds a record efficiency of 13.85% using Zn-Cu-In-Se NCs [4]. The classical synthesis of CIS NCs in organic media using long aliphatic ligands allows for a good control of their size dispersion, as well as compositional uniformity [5]. This is important for light emission applications such as Light-Emitting Diodes (LEDs) and bioimaging [6,7]. For photovoltaic applications size control is less critical, as in principle for CIS based absorbers ideally no quantum confinement exists to absorb most of the solar spectrum. However, organic synthesis of CIS NCs has several drawbacks, such as the use of solvents, high temperatures, requirement of inert conditions, and most importantly, the inevitable presence of long alkyl chain surface ligands to ensure steric stabilization of the colloids. These ligands considerably impact electron transfer processes from the NCs to the TiO_2_ photoelectrode in QDSSCs. To circumvent this problem, the initial ligands should be replaced by shorter ones prior to their utilization in QDSSCs. This step generally involves switching from steric stabilization to electrostatic repulsion, achieved for example by transferring the NCs from the organic to the aqueous phase using mercaptocarboxylic acids. However, besides being chemical- and time-consuming, post-synthetic ligand exchange is often accompanied by the generation of surface-related defects, which can act as trap states for photogenerated charge carriers. Moreover, typically used 1-dodecanethiol (DDT) ligands are very hard to replace because of their high affinity for the CIS core and the formation of a tight ligand double layer [8]. Direct synthesis of CIS NCs in aqueous medium is, therefore, a very appealing strategy. Aqueous synthesis also addresses ecological (avoiding toxic solvents) and economic (low temperatures, low cost) concerns related to NC syntheses in organics. Several methods have been proposed for CIS NCs [9], relying on classical heat-up approaches, using ambient pressure and standard heating [10,11,12,13,14], microwave heating [15,16,17,18,19], or hydrothermal [20,21,22,23,24,25] syntheses (Appendix A). The latter presents several advantages, such as the possibility to achieve higher temperatures yielding improved solubility of precursors and reaction products. Furthermore, hydrothermal conditions favor a better crystallization of the particles, yielding less surface defects and better luminescence, even without the addition of a shell, usually necessary in other procedures [26]. Several examples of CIS nanocrystals synthesized in water and used in QDSSCs exist [10,17,19,27,28,29] with a current record efficiency of 8.15% (Appendix A) [14].

A drawback reported for the aqueous synthesis is that CIS NCs are obtained with less well-defined size distribution and compositional control and hence optoelectronic properties. As an example, Xu et al. reported the hydrothermal synthesis of CIS NCs coated with glutathione having several emission bands in the photoluminescence (PL) spectrum [23]. These were attributed to the coexistence of several distinct NC species in solution, even though they have not been isolated. Size fractionation is a powerful tool to separate and study different families of NCs occurring in solution. It has been previously performed for CIS NCs synthesized in organic solvents using size-selective precipitation (SSP) [30,31,32,33]. More recently, Stroyuk et al. applied this method to study the optical properties of differently sized aqueous synthesized Ag-In-S/ZnS NCs [34].

In the present work, we hydrothermally synthesize 3-mercaptopropionic acid (MPA)-capped CIS NCs and apply SSP to separate them into three distinct fractions, emitting in the blue (413 nm), green (537 nm) and red (656 nm) spectral range. Compositional analysis and photophysical studies reveal that the blue fraction consists of strongly luminescent carbon nanodots, while the other two fractions correspond to CIS NCs of different size and composition. Upon sensitization of mesoporous TiO_2_ and ZnO nanowire electrodes, we demonstrate that panchromatic absorbing QDSSCs, i.e., using the non-fractionated sample, exhibit at a significantly higher power conversion efficiency than cells using either the individual green or red CIS NC fractions or their combination as absorber.

## 2. Materials and Methods

### 2.1. Materials

Copper (II) chloride dihydrate (99%), indium (III) chloride tetrahydrate (97%) and 3-mercaptopropionic acid (99%) were obtained from Sigma-Aldrich (São Paulo, Brazil). Thiourea (99%) was acquired from Dinâmica (São Paolo, Brazil). All chemicals were used without additional purification. Milli-Q ultrapure water was employed in the synthesis of the nanocrystals.

### 2.2. CIS NCs Synthesis

Aqueous copper indium sulfide NCs were synthesized using a hydrothermal method that was inspired by a published method [20]. In a standard procedure, 0.75 mmol of copper chloride and indium chloride were dissolved in 50 mL of deionized water, then 9 mmol of the surface stabilizer (MPA) were injected, and the pH value of the solution was adjusted to 10, using 3 M NaOH solution. Then, 1.5 mmol of CS(NH_2_)_2_ was added at room temperature. Other ratios have been also tested as indicated in the text. The prepared solution was transferred to a Teflon-lined stainless-steel autoclave, heated to various temperatures for different times as indicated and then left to cool down to room temperature. The obtained product was centrifuged and filtered to remove a small amount of insoluble aggregated matter. For SSP, first the as-prepared colloidal solution was concentrated in a rotary evaporator (to half of the initial volume), then a small portion (roughly 1 mL) of acetone was added to precipitate a first fraction of the material, which was separated by centrifugation. Then, another amount of acetone was added until the solution became cloudy in order to separate the second fraction, which was then also separated by centrifugation. A third fraction of the material remained in the supernatant. The three fractions were dried at 60 °C in vacuum.

### 2.3. Characterization

UV-Vis absorption and fluorescence spectra were acquired on a Cary 50 spectrophotometer (Agilent Technologies, Santa Clara, CA, USA) and on a RF-5301 PC spectrofluorophotometer (Shimadzu, Columbia, MD, USA), respectively. The absorption of TiO_2_ films was recorded using a Lambda 950 spectrometer equipped with an integration sphere (Perkin-Elmer, Waltham, MA, USA). The luminescence quantum yields were calculated by the comparative method using quinine sulfate (QY of 54%) as a reference. X-ray diffraction (XRD) patterns of the CIS NC samples on a non-reflective silicon holder and registered by a XRD-6000 diffractometer (Shimadzu, Columbia, MD, USA) operating in the scanning mode and using CuKα radiation (λ = 1.5418 Å) generated at 40 kV and at a current of 30 mA. Steps of 0.02° and a sampling time of 4 h were employed. Divergence slits of 1.0 mm, scattering slits of 1.0 mm, and receiving slits of 0.3 mm were used. Fourier-transform infrared spectroscopy (FT-IR) was conducted on a Spectrum GX spectrometer (Perkin-Elmer, Waltham, MA, USA). The spectra were obtained in the transmission mode in the range of 400 to 4000 cm^−1^ with accumulation of 32 spectra with the resolution of 2 cm^−1^. Transmission electron microscopy (TEM) was performed on a JEM 2100 FEG-TEM operating (JEOL BRASIL Instrumentos Científicos Ltd.a, São Paulo, Brazil) at 200 kV. The grid used was 300-mesh Lacey Formvar with ultrathin carbon film. The size of the nanoparticles was estimated using the ImageJ software (National Institutes of Health and the Laboratory for Optical and Computational Instrumentation, Bethesda, MD, USA). The zeta potential (ξ) of the aqueous dispersions was determined using a Delsa Nano 2.31 (Beckman Coulter, Brea, CA, USA). Scanning electron microscopy (SEM) images were acquired with a Zeiss Ultra 55+ Electron Microscope equipped with EDX detector (Zeiss, Jena, Germany). Electrochemical impedance analysis was performed using Autolab PGSTAT32 potentiostat (Metrohm AG, Herisau, Switzerland), over the bias potential range of 0.3 to 0.6 V and 20 mV of amplitude over the frequency range of 100 kHz to 0.1 Hz in dark condition. The Nyquist plots were fitted using Nova software (Metrohm Autolab, Herisau, Switzerland) according to a standard fitting model (see Supporting information). Ambient Pressure Photoemission Spectroscopy (APS) was performed using AC-3 (Riken Keiki, Tokyo, Japan).

### 2.4. NCs Deposition

The NCs were deposited on ZnO nanowires or on substrates covered with mesoporous TiO_2_ (10 ± 4 µm thick on glass/ fluorine-doped tin oxide-coated (FTO), 0.36 cm^2^, Solaronix (Aubonne, Switzeland)) by physisorption: the substrates were immersed in dispersions of NCs for 72 h followed by water and ethanol rinsing and applying ZnS as a blocking layer. For the latter, a SILAR deposition was used with a following cycle: immersion in 0.1 M aqueous Zn(NO_3_)_2_ for 60 s followed by rinsing and immersion into 0.1 M aqueous Na_2_S for 60 sec and rinsing. The cycle was repeated 3 times.

### 2.5. Solar Cells

NC sensitized solar cells were prepared using activated brass plates (treated by conc. HCl at 85 °C for 1 h) as counter-electrodes, polysulfide electrolyte (1 M Na_2_S and 2 M S in water) and Parafilm spacers. For the impedance measurements the cells were constructed using CuS counter-electrodes deposited by chemical bath deposited (CBD) on FTO, according to [35] and a Surlyn spacer using hot pressing. The solar cells were characterized under simulated sunlight (AM 1.5 G) at 100 mW/cm^2^ generated by a Newport class AAA solar simulator to test the photocurrent generation and energy conversion efficiency, the device active area was 0.36 cm^2^.

## 3. Results and Discussion

### 3.1. Influence of the Synthesis Parameters on the Properties of Aqueous NCs

Understanding and controlling the influence of various synthesis parameters on the NCs properties is of paramount interest in view of their optimization for optoelectronic applications, such as solar cells. First, the role of the synthesis duration was studied as at extended reaction times CIS NCs can grow uncontrollably in aqueous solutions leading to poor crystallinity. As a function of reaction time (5, 10 and 20 h), the absorption spectra of the resulting MPA-capped CIS NCs have onsets at 500-550 nm and feature a single peak, which shifts to lower energy with increasing synthesis time (Figure 1a, Appendix A). Typically, in the case of classical binary nanocrystals, it is referred to as the first excitonic peak and can serve as an indicator of particle growth as a direct consequence of reduced quantum confinement. In the case of ternary systems this peak is usually not observed due to defect-related states localized in the band gap, compositional dispersion and/or size distribution [36]. Recently, Jara et al. claimed that in the case of copper-deficient CIS NCs synthesized in organic medium an excitonic absorption peak was observed [37]. In our case, the NCs are also copper-deficient, and we observe comparably well-defined absorption peaks with a position depending on the reaction time. This behavior could be consistent with quantum confinement induced changes of the band gap of the NCs as a result of their growth. However, compositional changes during prolonged heating times cannot be excluded at this stage.

The obtained PL band shapes vary significantly and shift to longer wavelengths when increasing the synthesis duration (Figure 1b). Interestingly, at all reaction times studied, the PL spectra contain peaks at well-defined wavelengths (450, 520 and 650–680 nm). Their ratio changes with the reaction time, while their position remains almost the same, with the exception of the low energy peak at 650–680 nm, which gradually shifts to longer wavelengths. Consequently, at least three relatively stable emissive species are expected to be present in the colloidal solution.

Next, the influence of the temperature and pressure during the hydrothermal synthesis was investigated. At 125 °C (1.8 bar), 150 °C (3.0 bar) and 175 °C (6.8 bar) the PL spectra exhibit two distinct peaks in the red and green regions (Figure 1c,d). After 20 h of heating at 125 °C the emission is dominated by a higher energy peak (at 524 nm), while at 150 °C the main peak was in the red region (683 nm) accompanied by a minor secondary one at 507 nm. At 175 °C the two peaks appear already after 5 h of synthesis, and at longer times (20 h) precipitation occurs. Heating to 100 °C for 20 h did not result in the formation of NCs, as judged by the optical spectra, due to the low reactivity of the metal and sulfur precursors at this temperature [25]. Based on these observations, we can conclude that increasing the temperature influences the reaction rate, thereby, accelerating the formation of larger nanocrystals with a characteristic PL peak at ca. 670 nm. At the same time, the different reaction temperatures did not significantly change the reaction mechanism and resulted in NCs of similar size and composition, as judged by the absence of additional PL peaks and only slight shifts in the existing ones. At high temperatures and long reaction times, degradation of the surface ligands probably results in the aggregation of the nanocrystals and their subsequent precipitation.

In addition to the role of temperature and synthesis time, the influence of the copper precursor ratio was studied. Therefore, several copper concentrations were used in the synthesis: stoichiometric, Cu-deficient and in Cu excess (Cu-In-S 1:1:2, 0.5:1:2, 2:1:2) ratios. An increase of the initial copper concentration leads to NCs emitting exclusively in the blue/green region, whereas its reduction results in a batch containing several luminescent fractions (Appendix A).

### 3.2. Fractionation and Characterization of NCs

We expect that the distinct PL peaks observed using various conditions correspond to different families of CIS NCs with specific properties present as a mixture after synthesis. To verify this hypothesis, the product of the synthesis using a Cu:In:S reactants ratio of 0.5:1:2, a temperature of 150 °C and a reaction time of 20 h was fractionated by means of SSP. First, the absorption and emission spectra of the unfractionated mixture was measured (Figure 2). The PL spectrum of the unfractionated solution features a main peak at 530 nm. The valence band position was determined by ambient pressure photoemission spectroscopy (APS) at −5.39 eV. By adding the optical band gap calculated from PL we can determine the conduction band position as −3.05 eV, which is perfectly situated for the charge injection into the TiO_2_ photoelectrode in QDSSCs.

Then, the reaction mixture was concentrated and a small volume of anti-solvent (acetone) was added to the aqueous solution of NCs causing the formation of a precipitate, which was isolated and redispersed (CIS-1). Further acetone was added to the supernatant until a new precipitate was formed and separated (CIS-2). Finally, the remaining solution is referred to as CIS-3. The absorption spectra of the different fractions vary significantly, indicating that the fractionation procedure was successful: CIS-1 features an absorption onset at 650 nm, CIS-2 at 500 nm, and CIS-3 in the UV region at around 270 nm (Figure 2a). Likewise, distinctly different emission characteristics are observed in the PL spectra upon excitation at 355 nm (Figure 3b, Appendix A) with single PL peaks at 532 and 413 nm for CIS-2 and CIS-3, respectively, compared to the multipeak PL of unfractionated reaction mixture (Figure 1b). In the PL spectrum of CIS-1 two peaks can be distinguished: one similar to the fraction CIS-2 (537 nm) and another one of higher intensity in the red region at 656 nm. These results indicate that the fractions CIS-2 and CIS-3 have uniform compositions, while CIS-1 occurs as a mixture of two emissive species, CIS-2-like and another one. As for the absorption, variations of the PL spectra can originate from the difference in chemical composition and/or size of the fractionated NCs. Considering that the three fractions have been obtained by SSP, it can be concluded that they exhibit different solubility, which can be ascribed to different sized particles and/or to particles with different surface chemistry. The peak in the red-NIR region observed for the fraction CIS-1 can be likely ascribed to a radiative recombination, typically observed in CIS nanocrystals. It is now well-established that this transition involves photo-excited electrons delocalized in conduction band levels and holes localized in Cu-related intra-bandgap states (e.g., Cu vacancies) leading to radiative recombination with sub-bandgap energy [9,38]. To the contrary, the origin of the emission at lower wavelength (535 nm) is much less studied. Previously, similar peaks were observed in hydrothermally synthesized CIS/glutathione NCs [23] and in the heat-up aqueous synthesis of CIS/MPA NCs [39], albeit, showing a much broader line width. Our own interpretation, which is subject of ongoing research, for explaining the observed relatively sharp emission feature (Full width at half maximum, FWHM, 60–70 nm) implies the involvement of localized donor states close to the conduction band, leading to donor-acceptor pair recombination (Figure 2c).

Next, the isolated CIS fractions were studied by energy dispersive X-ray spectroscopy (EDX) in order to establish their elemental composition (Table 1, Appendix A).

The fractions CIS-1 and CIS-2 are strongly copper-deficient with Cu:In ratios equal to 1:5.1 and 1:7.4, respectively. Noteworthy, one of the advantages of ternary CIS materials is that they can tolerate large off-stoichiometries, which makes it possible to tailor their optoelectronic properties. Additionally, they have the ability to form a series of stable “ordered defect compounds” (ODCs) [40], one of which being CuIn_5_S_8_, corresponding well to the composition of CIS-1. Such copper deficiency is known to induce p-type self-doping in similar NCs and can improve the overall material conductivity [3]. Moreover, copper-poor colloidal CIS NCs were previously shown to improve the PL quantum yield (QY) [37,38,41,42,43,44] and the efficiencies of QDSSCs [14,45]. Here, the QYs for CIS-1 and CIS-2 were measured as 0.3%, and 1%, respectively. Of course, the QY of CIS NCs can be significantly improved by growing a passivating ZnS shell on their surface, which is beyond the scope of the present work. Zeta potential measurements (Appendix A) confirmed the negative surface charge of the NCs due to deprotonated MPA ligands supporting the colloidal stability.

In the fraction CIS-3 very few copper and almost no traces of In are found, while the concentrations of carbon, sulfur, and oxygen are the highest among all fractions. This suggests the presence of side-products, such as elemental sulfur and/or unreacted precursors remaining in solution. At the same time, CIS-3 has the highest QY among all fractions, 9.6%, and the PL wavelength is strongly excitation-wavelength dependent (Appendix A). These features, together with the very high C content determined from EDX, indicate the presence of luminescent carbon nanodots. The latter can be commonly synthesized using hydrothermal methods and typically emit in the blue/green region [46]. As an example, the observed PL spectrum of CIS-3 resembles closely that of carbon dots synthesized hydrothermally from folic acid as the starting compound [47]. In the present case, we propose that thermal degradation of MPA under the hydrothermal conditions induces the formation of carbon nanodots. Transmission electron microscopy (TEM) images of the samples results in mean diameters of the NCs of 4.5 ± 0.5 nm (CIS-1), and 2.6 ± 0.3 nm (CIS-2), respectively (Figure 3). It was not possible to obtain exploitable TEM images of the carbon nanodots, due to their expectedly small size < 2 nm and low contrast on the TEM grids, containing an amorphous carbon film of a few nanometers thickness.

In order to obtain more information about the structural features of the CIS NCs in the fractions 1 and 2, FTIR spectroscopy, powder X-ray diffraction (XRD), and TEM were used. The FTIR spectra of the two CIS fractions reveal characteristic bands of the MPA ligand corresponding to the vibrations of its functional groups, at 3300–3400 cm^−1^ (hydroxyl groups) as well as at 1550–1570 and 1395–1405 cm^−1^ (carboxylate, Figure 4a, Appendix A). The signal of the stretching vibration of the thiol S–H group, occurring in the 2550–2680 cm^−1^ region, is absent, which is expected: at pH 10 used during synthesis, the thiol groups are deprotonated and the resulting thiolate interacts easily with soft Lewis acids, such as copper cations at the NC surface.

Powder X-ray diffraction shows that the fractions CIS-1 and CIS-2 are characterized by broad peaks corresponding to small-sized crystallites (Figure 4b). The peaks at 2Θ = 28°, 47° and 55° observed for CIS-1, which are less defined for CIS-2, match well the 112, 204/220, and 116/312 reflections of chalcopyrite CuInS_2_ (JCPDS # 85-1575) [20]. HRTEM images corroborate this identification showing the presence of NCs with 3.1 Å lattice spacings corresponding to the (112) facet of chalcopyrite CuInS_2_ in the CIS-1 fraction (Figure 3a). Nonetheless, a possible contribution of a copper-poor phase like Cu_0.74_In_4.96_S_8_ (JCPDS 04-001-6679) cannot be excluded in view of the data quality. The hump visible in both diffractograms at around 2Θ = 20° has been observed in many other NC syntheses and is attributed to organic residues. The XRD pattern of the CIS-3 fraction has a single broad peak at 2Θ = 22°, typical for carbon nanodots (Appendix A).

Summarizing the optical, morphological and chemical studies of the fractionated CIS NCs, we can identify the major species formed during hydrothermal synthesis. The first fraction (CIS-1) consists of a copper-poor Cu-In-S NC batch (Cu:In = 1:5) of around 4.5 nm size, emitting in the red (600–700 nm) spectral region. This emission is ascribed to a recombination mechanism implying transitions from photoexcited electrons delocalized in the conduction band to localized Cu-related acceptor states close to the valence band [48]. Consequently, the PL peak shifts as a function of synthesis time, temperature, and chemical composition. The second fraction (CIS-2) consists of smaller (2.5 nm) and more copper-deficient Cu-In-S NCs (Cu:In = 1:7.4) with a PL maximum in the green region (530–540 nm). This emission is almost independent of the synthesis conditions, as would be the case in transitions between well-defined donor and acceptor states described by the donor-acceptor pair (DAP) recombination mechanism [49]. Finally, carbon nanodots with a PL in the UV-blue region are formed from the organic precursors (MPA, thiourea) as a side-product under the high pressures developed under hydrothermal conditions. These three species are always produced with the presented type of synthesis, while their relative ratios vary as a function of the reaction conditions.

### 3.3. Solar Cell Integration

To explore one possible application of the hydrothermally synthesized CIS NCs, they were used as sensitizers in QDSSCs. As one appealing feature they contain carboxylate functions on their surface, which can be directly used to interact with the surface of oxide-containing photoelectrodes. For solar cell preparation, first, activated mesoporous TiO_2_ electrodes were immersed into concentrated colloidal solutions of CIS NCs for a given time followed by rinsing and passivation coating with a thin ZnS layer obtained by SILAR deposition. The sensitized electrodes were then assembled with brass counter-electrodes using a thin spacer and a polysulfide electrolyte was injected into the cavity. The performance of the resulting solar cells was studied under simulated solar irradiation (Table 2, Figure 5).

Solar cells resulting from the non-fractionated sample used as-is (non-purified) after synthesis demonstrate a comparably high open-circuit voltage (V_OC_) of 0.53 V and short-circuit current density (J_SC_) of 13.62 mA/cm^2^ in average. Without the ZnS coating, the solar cells develop a decreased V_OC_ of around 0.4 V highlighting the importance of the passivation layer. It decreases undesired recombination processes of the electrons injected into TiO_2_ with oxidized NCs and the polysulfide electrolyte. The fill factor (FF) is also comparably high (around 0.6), characteristic of good interface quality. The overall photoconversion efficiency (PCE, η) is 4.53% (best cell: 4.67%), which is by far the highest value reached with hydrothermally synthesized ternary NCs in QDSSCs and on par with other aqueous QDs (Appendix A), demonstrating the highest fill factor ever achieved for similar materials (0.64) [1,2].

Next, we tested the stability of the cells fabricated because this parameter is rarely studied in reports on QDSSCs. The most efficient cells using, according to a widely applied procedure, a Cu*_x_*S counter-electrode directly prepared on brass had a limited stability of several hours due to electrolyte leakage and continuous brass corrosion by the polysulfide electrolyte. In the recent literature, CBD CuS electrodes were successfully used to improve the cells’ stability due to the direct deposition on FTO and thus better sealing and absence of corrosion [33]. Even though cells using CBD-CuS electrodes resulted in comparably lower overall performances (Appendix A), due to the lower photocatalytic activity than brass-derived Cu*_x_*S, the stability of the cells increased significantly, to at least three weeks (Appendix A).

The improved stability allowed for advanced solar cell characterization, using electrochemical impedance spectroscopy (EIS). This method allows studying the charge recombination and resistances at different interfaces within the solar cell (Appendix A). A further advantage of the CBD-CuS counter-electrodes was that the brass-based ones turned out to be sensitive towards the impedance polarisation probing. Notably, the electron lifetime (R_rec_ × C_µ_) achieved was very high (700–1700 ms), indicating a slow recombination rate in such hydrothermally synthesized NCs compared to previously studied CIS QDs synthesized in organics (Appendix A).

Subsequently, to evaluate the effect of the structure and composition of the CIS NCs on the photovoltaic properties of QDSSCs, the fractionated NCs solutions were used as sensitizers. Despite a significant difference in the optical properties, both CIS-1 and CIS-2 fractions display similar photovoltaic behavior. CIS-2 demonstrates slightly better J_SC_ and V_OC_ characteristics, compared to CIS-1, probably due to a more efficient charge injection. Its PL QY is higher, which can be interpreted as an indicator of reduced non-radiative recombination pathways (Figure 5). The average PCE of both hydrothermally synthesized CIS NC fractions is 2.52% (CIS-1) and 2.85% (CIS-2), respectively, which is similar to most aqueous QDSSCs reported. These results clearly show that each CIS fraction taken separately yields lower performance as compared to the unfractionated NC solution. An important parameter contributing to this behavior is the lower photocurrent observed in the case of the individual CIS NC fractions, which could be related to less efficient light harvesting and/or charge transfer/extraction. The former can be explained by a lower spectral absorption range of the fractionated samples and/or their lower loading on TiO_2_. However, a comparison of the optical absorption spectra of TiO_2_/QD films show no significant improvement of the absorbance of the unfractionated sample with respect to the individual CIS-1 and CIS-2 fractions (Appendix A). Therefore, the most likely explanation for the higher photocurrent in the cells based on unfractionated NCs is improved charge extraction from photo-excited unfractionated NCs, the reason of which is not yet understood.

Another factor, contributing to a lesser extent to the lower solar cell efficiency of the individual NC fractions, is the lower fill factor observed with the individual fractions, which is likely related to charge recombination within the cell. According to the EIS results, the recombination resistance using the unfractionated solution is significantly higher over the range of potentials (Appendix A), probably due to the excess of ligands in the unpurified solution, allowing for the better NC surface passivation, and thus, decreased back charge transfer with the electrolyte.

Due to their size- and composition dependent absorption spectra, NC absorbers provide another appealing way to improve the solar cell performance by optimizing the light harvesting and charge collection efficiency via so-called quantum funneling [50]. In this approach NC layers with graded band gaps are assembled in a way to better drive the excitons towards the electrodes. This concept is particularly used in NC thin film solar cells, even though Kamat et al. also demonstrated its relevance for QDSSCs using CdSeS NCs [51]. The fractionation of NC solutions can be an efficient way to generate NCs with gradually varying band gaps adapted for quantum funneling. To test this concept with the present samples, we prepared bilayer devices using CIS-1 and CIS-2 fractions with two objectives: (i) to harvest more light due to the complementarity in their absorption spectra, and (ii) to better collect the photogenerated charges, due to preferential electron transfer from higher band gap CIS-2 towards lower gap CIS-1 and then to TiO_2_. For this purpose, the electrodes were first sensitized with CIS-1, followed by CIS-2. The resulting cells indeed demonstrate an increased conversion efficiency (2.91%, best cell: 3.19%) than those obtained with the individual fractions, however, the expected increase in J_SC_ is modest (Table 2). One possible reason could be different NC loading: compared to the previous studies, where different NC batches were deposited on TiO_2_ by electrophoresis [51], in our case, a simple consecutive chemisorption is probably less efficient in building a robust bilayer for efficient quantum funneling.

For the fractionated samples and their combinations, the PCE of the resulting QDSSCs are significantly lower than that of the initial sample containing all three fractions. The main reason is the lower short-circuit current density, which is related to less efficient light absorption and charge extraction.

Finally, we also attempted to use ZnO nanowires as nanostructured electrodes for sensitization with CIS NCs. Due to their vertical orientation on the substrate, ZnO nanowires provide in principle excellent channels for efficient electron transfer to the FTO electrode [52]. However, the observed photovoltaic performances (Appendix A, Appendix A) were much lower, compared to TiO_2_. Judging from the coloration of the electrode, this is probably a consequence of lower NC loading compared to similar systems using ZnO nanowires sensitized with CIS NCs prepared in organic phase [53]. Despite the lower efficiencies, the general trend observed for mesoporous TiO_2_ sensitized electrodes is also valid for ZnO nanowires: the efficiencies of the fractionated CIS NCs are lower than those of the unpurified samples, mainly due to a much lower photocurrent.

## 4. Conclusions

In conclusion, hydrothermal synthesis is demonstrated to be a method with a high potential for the simple, cost-efficient and eco-friendly fabrication of aqueous CuInS_2_ NCs suitable for light emission and photovoltaic applications. The present approach gives direct access to a mixture of red-, green- and blue-emitting NCs, which can be conveniently separated using size-selective fractionation. While the red and green fractions correspond to copper-poor CIS NCs of different size and composition, the third fraction consists of carbon nanodots emitting in the blue region with a high quantum yield (around 10%), formed as an unexpected side-product of the reaction. When used to sensitize mesoporous TiO_2_ electrodes of QDSSCs, the obtained NCs resulted in power conversion efficiencies of up to 4.67% with a V_OC_ above 0.5 V and a FF above 0.6. These values constitute the highest obtained so far for QDSSCs applying hydrothermally synthesized CIS NCs. The use of the isolated, purified CIS fractions leads to a decrease of the solar cell performances, in particular of the photocurrent. These results indicate that the two CIS NC fractions and the carbon nanodot fraction lead to synergistic panchromatic absorption. The separation and identification of the different hydrothermal synthesis products paves the way for the optimization of CIS NCs to obtain tailored properties, leading to “green” QD-sensitized solar cells, with improved performance. Another possible application of the RGB emitting fractions could be in white light generation or in displays, in particular after enhancing their PL QY, e.g., through surface engineering.

## Figures and Tables

**Figure 1 nanomaterials-10-01252-f001:**
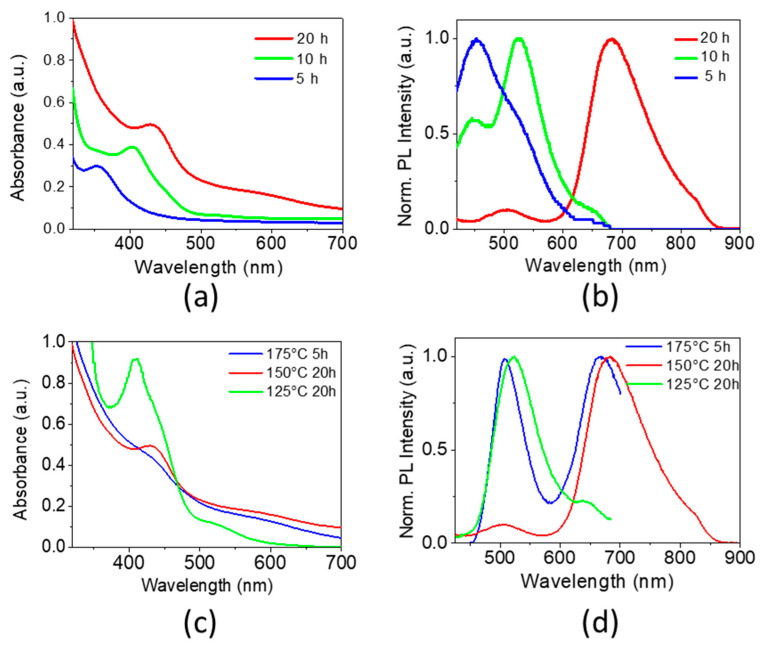
Absorbance (**a**,**c**) and normalized photoluminescence (PL) (**b**,**d**, excitation wavelength: 355 nm) spectra of copper indium sulfide (CIS) nanocrystals (NCs) prepared using different synthesis times at 150 °C (Cu-In-S ratio = 1:1:2) (**a**,**b**), and using different synthesis temperatures (**c**,**d**). At 175 °C for longer reaction times precipitation occurs.

**Figure 2 nanomaterials-10-01252-f002:**
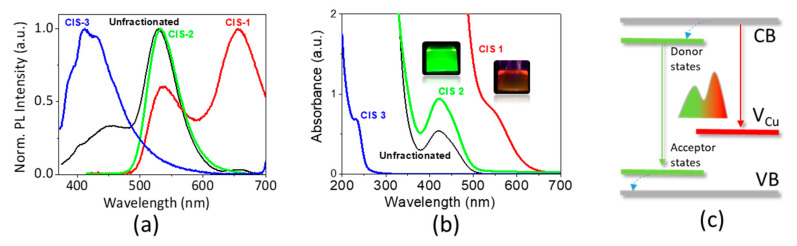
Absorption (**a**) and emission (**b**) spectra of the different CIS NC fractions obtained by size-selective precipitation (SSP) (Cu-In-S = 0.5:1:2, T = 150 °C, t = 20 h). (**c**) Scheme of the hypothetic radiative recombination pathways at the origin of the PL emission in CIS-1 (red) and CIS-2 (green).

**Figure 3 nanomaterials-10-01252-f003:**
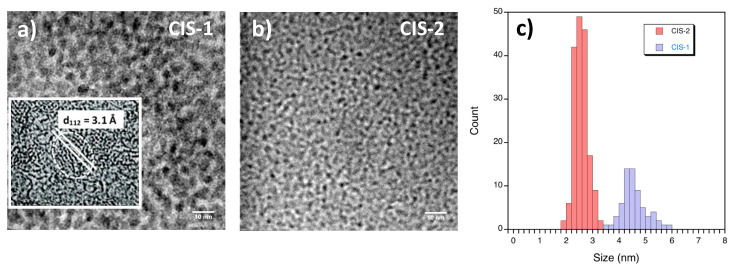
TEM images of CIS-1 (with a corresponding high resolution TEM (HRTEM) image as inset) (**a**) and CIS-2 (**b**) NCs, and their size distribution (**c**).

**Figure 4 nanomaterials-10-01252-f004:**
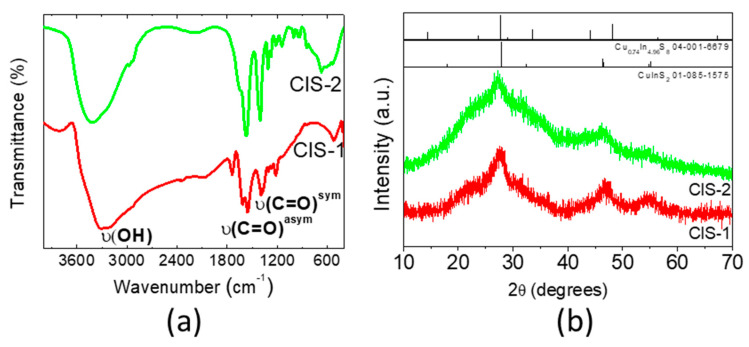
Infrared (IR) spectra (**a**) and X-ray diffractograms (**b**) of the fractions CIS-1 and CIS-2.

**Figure 5 nanomaterials-10-01252-f005:**
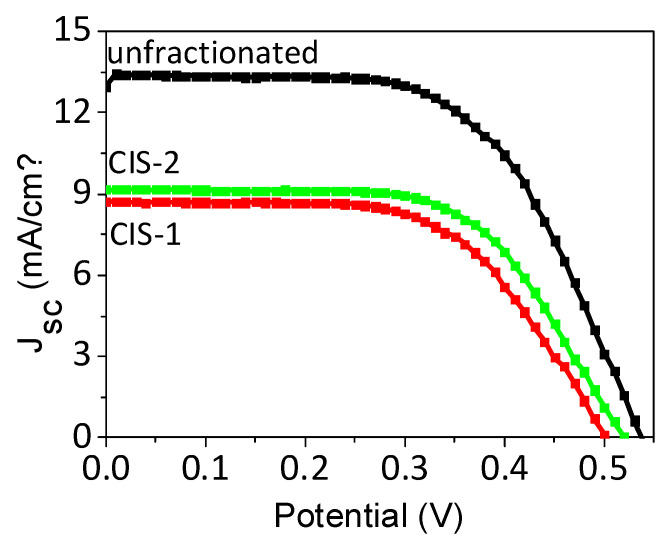
Photovoltaic behavior of solar cells sensitized with various fractions of CIS NCs.

**Table 1 nanomaterials-10-01252-t001:** Elemental composition of CIS fractions obtained by EDX. The values are average of 5 measurements and were normalized to the copper concentration.

Fraction	Cu	In	S
CIS-1	1.00	5.06 ± 0.79	8.24 ± 1.42
CIS-2	1.00	7.40 ± 2.86	38.05 ± 12.29
CIS-3	1.00	0.02 ± 0.03	56.70 ± 5.61

**Table 2 nanomaterials-10-01252-t002:** Photovoltaic parameters of TiO_2_ based solar cells sensitized by CIS NCs. The champion cell and average parameters of three cells for each condition are given.

Sample	Cell	V_OC_ (V)	J_SC_ (mA/cm^2^)	FF (%)	η (%)
CIS 1	champion cell	0.50	8.73	59	2.60
	average	0.51 ± 0.01	8.53 ± 0.30	58 ± 1	2.52 ± 0.11
CIS 2	champion cell	0.52	9.18	61	2.91
	average	0.52 ± 0.01	9.04 ± 0.12	60 ± 1	2.85 ± 0.07
CIS-1 + CIS-2	champion cell	0.53	9.09	67	3.19
	average	0.52 ± 0.01	8.68 ± 0.58	65 ± 4	2.91 ± 0.40
Unfractionated CIS	champion cell	0.52	13.96	64	4.67
	average	0.53 ± 0.11	13.62 ± 0.32	63 ± 1	4.53 ± 0.12

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
