# Peer review of "Hydrothermal Synthesis of Aqueous-Soluble Copper Indium Sulfide Nanocrystals and Their Use in Quantum Dot Sensitized Solar Cells"

_nanomaterials, 2020, doi:10.3390/nano10071252_

Round 1

Reviewer 1 Report

The authors reported that ternary copper indium sulfide nanocrystals (CIS-NCs) are synthesized using a hydrothermal method by varying the synthetic conditions such as aging time, temperature, pressure, etc. and dispersed in aqueous medium by stabilizing the short-chain polar solvents. In addition, these are utilized in the quantum dot-sensitized solar cells (QDSSc) showing the improved photovoltaic performance by mixing the CIS-NCs with various size due to panchromatic absorption. For this approach, they choose the size-selection precipitations from the as-synthesized CIS-NCs. I recommend that this manuscript can be acceptable to the publication in Nanomaterials after minor revision as follows:

  1. They mentioned that a record efficiency of CIS-NC-based QDSSc is 8.15%, but the efficiency in this study is 4.67%. Please discuss why your device efficiency is lowered that a record device.
  2. In supporting information, they showed the photo images of resultant CIS-NCs in solution-phase. I think CIS-NCs can be considered as a light emitting semiconductor. Please search the CIS-NC-based light-emitting device (LED) and add the literatures in introduction part if possible.
  3. In main manuscript, there are UV-Vis absorption, PL, FT-IR, XRD spectra as main figures. I recommend that photo and TEM images should include in the manuscript as a main figure. In addition, please rearrange the figures for making the author`s claims stronger.

Author Response

We would like to thank the reviewer for the positive evaluation of the manuscript.

  1. They mentioned that a record efficiency of CIS-NC-based QDSSc is 8.15%, but the efficiency in this study is 4.67%. Please discuss why your device efficiency is lowered that a record device.

 Indeed, our performance is below the reported record in the field. However, several points need to be taken into account. First, the record (8.15%) published has been obtained using a non-standard light intensity of 0.3 Sun [RSC Adv. 2016, 6, 100145], which has definitely influenced the end result and we ignore what it would have been with the standard 1 Sun irradiation power. In our work we used a calibrated 1 Sun AM1.5G AAA solar simulator. Second, the synthesis method of CIS QDs in this work is different, hydrothermal compared to heating up, which may lead to different QD sizes and compositions. Third, the aim of this manuscript was not to fabricate a record-breaking solar cell by optimizing all the parameters (QDs shelling, electrode thickness, loading, electrolyte, counter-electrode…), but rather to demonstrate that the obtained hydrothermally synthesized CIS QDs can be successfully used as sensitizers in QDSSCs, and to show the influence of their fractionation on the performance of the cells. Finally, among CIS QDs synthesized using similar procedures, our results correspond to the highest efficiency. 

  1. In supporting information, they showed the photo images of resultant CIS-NCs in solution-phase. I think CIS-NCs can be considered as a light emitting semiconductor. Please search the CIS-NC-based light-emitting device (LED) and add the literatures in introduction part if possible.

 We would like to thank the reviewer for the suggestion. Indeed, such aqueous-based QDs could be used in light-emitting diodes and the perspective proposed is interesting and appealing.

Actions taken: References 5 (Wang, Z.; Zhang, X.; Xin, W.; Yao, D.; Liu, Y.; Zhang, L.; Liu, W.; Zhang, W.; Zheng, W.; Yang, B.; et al. Facile Synthesis of Cu-In-S/ZnS Core/Shell Quantum Dots in 1-Dodecanethiol for Efficient Light-Emitting Diodes with an External Quantum Efficiency of 7.8%. Chem. Mater. 2018, 30, 8939–8947) and 6 (Chuang, P.; Lin, C.C.; Liu, R. Emission-Tunable CuInS2/ZnS Quantum Dots: Structure, Optical Properties, and Application in White Light-Emitting Diodes with High Color Rendering Index. ACS Appl. Mater. Interfaces 2014, 6, 1764–1769) added to the Introduction, p. 1, l. 37. Also, the expression “light emission” has been added in the conclusions (p. 10, l. 394).

  1. In main manuscript, there are UV-Vis absorption, PL, FT-IR, XRD spectra as main figures. I recommend that photo and TEM images should include in the manuscript as a main figure. In addition, please rearrange the figures for making the author`s claims stronger.

Actions taken: Following the suggestion of the reviewer we have regrouped the Figures 1 and 2 together as Figure 1, in addition we have moved the figure of TEM images from the supporting information to the main text as a new Figure 3.

Reviewer 2 Report

The authors demonstrate hydrothermal synthesis of copper indium sulfide nanocrystals and their application in quantum dot sensitized solar cells. The authors systematically investigate the reaction condition and characterize the optical properties of CIS QDs. And quantum dot sensitized solar cells are fabricated with different types of CIS QDs and their photovoltaic properties are characterized. However, there are many literature reporting the methods to synthesize CIS QDs and their applications for QD-sensitized solar cells and it is hard to find the novelty of this work for the publication. In addition, microscopic characterization of QDs should be performed.

Author Response

We thank the reviewer for the analysis of our manuscript. However, we strongly disagree with the judgement concerning the lack of novelty with respect to the state of the art. Indeed, several works exist on the aqueous synthesis of CIS NCs, however, here we focus on the hydrothermal synthesis, which has been very scarcely studied before and at the same time presents numerous advantages.

More specifically, we show that the observed differences in optical properties and PV performance of the obtained QDs are not directly related to quantum confinement (i.e. size) effects but rather to their stoichiometry and chemical nature. This is an important new insight gained from the investigation of size-selected fractions of the QDs. Furthermore, our optimized hydrothermal synthesis results in green and red-emiting CIS QD fractions of unprecedented narrow emission line width, which could be of high interest also for other applications than PV (e.g., LEDs as suggested by Reviewer 1). In addition, we demonstrate for the first time that hydrothermal synthesis does not only yield CIS QDs but can also lead to the simultaneous formation of carbon nanodots. The solar cell experiments clearly show that this “side-product” effectively contributes to the light absorption, charge carrier generation and hence photocurrent. Therefore, these results reveal an interesting novel strategy for the design of panchromatic absorbing solar cells based on different families of nanoparticles.

Finally, we point out that the developed QDs have a very high environmental potential as they do not contain toxic elements, nor organic solvents are used for their synthesis and cost-efficient procedures are applied for the fabrication of the solar cells.

The microscopic characterization of the QDs studied has been performed. Following the suggestion of Reviewer 1, we have added it as a Figure in the main text of the manuscript (Fig. 3). To visualize these QDs two different TEMs have been used, albeit their imaging remains challenging because of their small size and low contrast due to the low amount of high-Z elements.

Reviewer 3 Report

The authors have reported the synthesize of water-soluble copper indium sulfide (CIS) nanocrystal by facile hydrothermal method. The CIS materials was used as light absorbers in quantum dot sensitized solar cells. They have provided a couple of fundamental characterizations. It attracts the researchers working in this field.

Comments:

  1. Include the error in Table 1.
  2. Since XRD may not be so sensitive to detect the secondary phase as Raman spectra, it would be better if the synthesized CIS nanocrystal can be probed by Raman Spectra?
  3. Most of plots do not show unit in Y-axis. It is recommended to include.
  4. The observation of single and double PL characteristics peak is interesting. Is it possible to illustrated by schematic? It would be easy to the phenomena behind it.
  5. A minor suggestion, on page#2, line 69, it is better to split the paragraph……………In the present work, we hydrothermally synthesize 3-mercaptopropionic acid (MPA)-capped CIS NCs.

Author Response

We would like to thank the reviewer for the positive evaluation of the manuscript.

1. Include the error in Table 1.

Actions taken: the errors in the composition determined by EDX were added to the Table 1. 

2. Since XRD may not be so sensitive to detect the secondary phase as Raman spectra, it would be better if the synthesized CIS nanocrystal can be probed by Raman Spectra?

We would like to thank the referee for this suggestion, indeed Raman spectroscopy might be a useful method for the QDs phases determination. However, we are unfortunately not able to perform Raman measurements for this article as we do not have direct access to this equipment and access via a collaboration is problematic at this moment due to the current Covid restrictions.

3. Most of plots do not show unit in Y-axis. It is recommended to include.

We would like to thank the referee for this suggestion, indeed the units were sometimes missing.

Actions taken: the titles and units were added to all the graphs in the manuscript. 

4. The observation of single and double PL characteristics peak is interesting. Is it possible to illustrated by schematic? It would be easy to the phenomena behind it.

 Following this recommendation we included a scheme depicting the hypothetic recombination pathways occurring for the two different CIS fractions.

Actions taken: The corresponding scheme has been added as Figure 2C.

5. A minor suggestion, on page#2, line 69, it is better to split the paragraph……………In the present work, we hydrothermally synthesize 3-mercaptopropionic acid (MPA)-capped CIS NCs.

Actions taken: Following the suggestion, the corresponding paragraph has been split in two.

Round 2

Reviewer 2 Report

The author address all questions. Publication is recommended.